# Diversity Patterns of Epiphytic Orchids Along Elevation in the Mountains of Western Nepal

**DOI:** 10.3390/plants13223256

**Published:** 2024-11-20

**Authors:** Manisha Awasthi, Santosh Thapa, Bandana Awasthi, Chae Ryeong Lim, Young Han You, Ki Wha Chung

**Affiliations:** 1Department of Biological Sciences, Kongju National University, 56 Gongjudaehak-ro, Gongju 32588, Republic of Korea; manisha@smail.kongju.ac.kr (M.A.);; 2Department of Botany, Tri-Chandra Multiple Campus, Tribhuvan University, Ghantaghar, Kathmandu 44600, Nepal; thapasantosh42@gmail.com; 3Nepal Paani Program, Development Alternatives Incorporation, United States Agency for International Development, Panchakanya Marg, Kathmandu 44600, Nepal

**Keywords:** biodiversity, environment, epiphytic orchids, Himalayas, species richness

## Abstract

The biodiversity and distribution of epiphytic orchids are strongly influenced by their relationship with host plants, and environmental variables like elevation, slope, and local climate are key factors in determining the abundance and diversity of these orchids. The aim of this study was to examine the richness of orchid species at different elevations within the research area of Nepal. Sampling was conducted at elevations ranging from 1300 m to 2800 m above sea level, using a systematic sampling technique known as belt transects. Six circular plots with a diameter of 5.6 m were established in a horizontal transect at each site, spaced at least 50 m apart, with 100 elevation interval. The analysis revealed a hump-shaped relationship between orchid species richness and elevation, with the highest species richness observed at altitude of 2100–2200 m. The abundance of orchids was significantly correlated with host characteristics, including habit (shrub/tree), bark texture, nature (deciduous/evergreen), and physical factors. This study underscores the significant contribution of host characteristics and environmental factors in explaining the diversity of epiphytic orchid species along the elevation of the Himalayas.

## 1. Introduction

Western Himalayan region in Nepal is celebrated for its extraordinary biodiversity, including a plethora of unique plant species that thrive amidst its captivating landscapes [1,2]. Orchids are among the most fascinating taxa of this region’s flora due to their enchanting beauty and ecological roles [3]. The Orchidaceae family is a very large diversified group of flowering plants with worldwide distribution of around 28,000 species. The studies of these orchids, which are known for their medicinal properties and intricate relationships with their host trees, have provided insight into the complex interplay between species and their environments [4,5,6]. The elevation gradients of the Western Himalayas offer a unique natural laboratory for investigating how orchid species richness and community composition change with altitude, shedding light on broader ecological patterns and processes [7].

Orchids are largely classified into two groups: epiphytic and terrestrial. Many epiphytic orchid species that grow on trees rather than in the soil are residing in the Himalayas. The diversity of epiphytic species is influenced by a variety of factors, including the characteristics of host trees and the effects of environmental disturbances [2,8,9]. These studies have not only expanded our understanding of epiphyte-host dynamics but also highlighted the importance of epiphytes in ecosystem functioning, offering habitats and resources to numerous insects and microorganisms [10]. Moreover, the investigation into vascular epiphytes along altitudinal gradients has revealed a richness peak at mid-elevations, a pattern that prompts further exploration into the mechanisms driving epiphytic orchid diversity in the region [11,12].

In Nepal, the distribution of orchids, particularly terrestrial and epiphytic types, is influenced significantly by the country’s diverse topography and altitude gradient. Higher altitudes with cooler, humid microclimates favor terrestrial orchids, while lower, humid regions with dense canopy cover support epiphytic orchids that thrive in moist, filtered light conditions provided by their hosts. The relation between altitude and orchid diversity, showing that epiphytic orchids are commonly found at lower altitudes (500–1500 m), where warmth and moisture are abundant, while terrestrial orchids are more prevalent at mid-to-high altitudes. Altitude, canopy cover, and microclimate as key factors in orchid distribution, showing species-specific adaptations to different altitudinal ranges [4,7,9,13].

In the Kathmandu Valley, orchids face significant conservation challenges, primarily due to urbanization, deforestation, and habitat fragmentation. As the valley continues to urbanize, the natural habitats of orchids are increasingly threatened by infrastructure development and agricultural expansion [14]. Additionally, illegal collection of orchids for ornamental and medicinal purposes, along with climate change, has further exacerbated the vulnerability of these plants [15]. Conservation efforts in the region need to focus on habitat protection, sustainable management, and raising awareness about the ecological value of orchids to ensure their survival in this rapidly changing environment.

The impact of human activities on epiphytic orchids, particularly in regions like the Kathmandu Valley, underscores the urgency of integrating conservation efforts with the study of these species’ distribution and ecological roles [13,16]. Through a comprehensive understanding of the spatial patterns, drivers, and conservation needs of epiphytic orchids in the Western Himalaya, researchers can develop more effective strategies for preserving these vital components of Nepal’s biodiversity amidst environmental change [13,17].

This study was conducted to examine the diversity, species richness, and distribution of epiphytic orchids in the Khathkhola-3 area of Baglung and Beni-3 area of Myagdi of Nepal. The findings provide baseline data essential for the conservation of these orchids, thereby contributing to the preservation of Nepalese flora.

## 2. Results

### 2.1. Species Richness

Sampling of epiphytic orchids were archived in the areas of Kathekhola Rural Municipality-3 and Beni Municipality-3 with an elevation range of 1300–2800 m (Figure 1). Many epiphytic and terrestrial orchid species have been documented in the Tinchule mountain. This study observed a total of 79 epiphytic orchid species representing 25 genera across elevation ranges from 1300 m to 2800 m. They were found on 68 host species consisting of 30 different families (Appendix A). Some epiphytic orchid species observed during the field visit are shown in Figure 2.

The epiphytic orchid species richness (numbers) showed a bell-shaped distribution curve according to the elevation. The overall diversity of orchid species exhibited a range from 22 to 41 along the 100 m-elevation gradients, reaching its peak at 2100 m to 2200 m with the highest 41 species (Figure 3). The mean species richness was 32.56 ± 5.31.

Among the host families, Fagaceae was the most abundant for epiphytic orchids, accounting for 24.8% (553 species) followed by Lauraceae for 11.8% (263 species) and Ericaceae 10.1% (224 species), whereas least is seen in Taxaceae with 0.2% (5 species) (Figure 4). The families with higher abundance primarily consist of species with larger diameter, rough bark texture, evergreen, and greater height.

### 2.2. Ecological Factors and Their Interaction

We observed a significant increase in the richness of epiphytic orchid species with respect to the diameter at breast height (DBH) and the height of the host species. Firstly, the DBH of host species showed a strong positive correlation (*R*^2^ = 0.69) with the richness of epiphytic orchids (Figure 5A). Secondly, the height of the host species also demonstrated a significant positive association (*R*^2^ = 0.51) with the species richness (Figure 5B). Additionally, this study compared various ecological factors influencing the distribution and richness of orchid species. A significant difference was observed in species richness between deciduous (29.7 ± 4.5) and evergreen categories (33.8 ± 5.2), with evergreen host trees generally supporting a higher number of epiphytic orchid species than deciduous host tree species (*p* < 0.001, Figure 5C).

We analyzed the bark texture of the host species categorized as smooth, medium and rough and showed that orchid richness is significantly higher in rough bark texture (35.0 ± 4.5) such as *Quercus lamellosa, Quercus semecarpifolia, Rhododendron arboreum*, and *Schima wallichii* compared to both smooth such as *Alnus nepalensis* and *Ilex dipyrena* (28.9 ± 3.9) and medium bark textures such as *Ficus neriifolia*, *Juglans regia*, and *Aesculus indica* (30.1 ± 5.0), with statistical significances (*p* < 0.001). Medium bark showed slightly higher value than smooth bark, however, there is no significant difference between them (*p* = 0.366). This suggests that the rough bark texture’s performance or response is notably distinct in comparison to the other two bark textures, while the smooth and medium bark textures exhibit similar values for the measured variable (Figure 5D).

Further analysis reveals a curve in orchid species richness across vertical forest layers, highlighting significant differences in species richness between these layers (Figure 5E). The upper basal layer supports the highest number of epiphytic orchid species (34.4 ± 5.0), exhibiting significantly greater compared to the lower basal layer (31.9 ± 3.9, *p* = 0.006), the lower canopy layer (28.7 ± 4.5, *p* = 0.002), and the upper canopy layer (24.1 ± 2.8, *p* < 0.001). The lower basal layer, while having fewer species than the upper basal layer, still showed significantly higher species richness than the lower canopy layer and the upper canopy layer (*p* < 0.001). Similarly, the middle layer supports a higher number of species (32.0 ± 5.4) compared to the upper canopy layer (*p* = 0.006). These comparisons roughly revealed a bell-shaped richness in epiphytic orchids, with the highest distribution peak at the slightly lower levelled median height of the host trees.

Additionally, in the context of the land type where host species are found, the analysis revealed significant differences in species richness across various land-use types (Figure 5F). As expected, forest areas exhibited significantly higher species richness (32.6 ± 5.8) compared to agricultural lands (28.1 ± 2.3, *p* = 0.014) and settlement areas (25.9 ± 2.5, *p* = 0.015). Mixed-use areas, in turn, show significantly higher species richness (34.0 ± 4.2) compared to agricultural lands and settlement areas (*p* < 0.001). Overall, this study indicated that forest areas support the highest number of epiphytic orchid species, followed by mixed-use areas, agricultural land, and settlement areas. These findings underscore the impact of land-use type on the richness of epiphytic orchids, with natural forest areas providing the most favorable conditions for their diversity.

Pearson’s correlation test showed the relationship among elevation, DBH and the height of host trees (Table 1). Elevation has weak negative correlations DBH and height but with no significant differences (DBH: *r* = −0.13, *p* = 0.07; height: *r* = −0.06, *p* = 0.42). Conversely, DBH and height exhibit a strong positive correlation (*r* = 0.74, *p* < 0.001).

Elevation, a macro-environmental variable, accounted for 24% of the total dataset’s variance. Similarly, other micro-variables such as DBH, rugosity and height of the host species exerted significant influences on the formation of heterogeneous species composition. The spatial distance between the sample points in Figure 6 approximated the dissimilarity in species composition.

*Pleione humilis*, *Pinalia graminifolia*, *Dendrobium monticola* and *Liparis resupinata* had strong affinity towards higher elevation and distributed toward the positive end of canonical correspondence analysis (CCA) axis 1 with host species like *Quercus lamellose*, *Quercus semecarpifolia*, and *Tsuga dumosa* (Figure 6). The negative end of CCA axis 1 and axis 2 explained that the species like *Agrostophyllum callosum*, *Bulbophyllum odoratissimum*, *Bulbophyllum umbellatum*, *Coelogyne fimbriata*, *Coelogyne prolifera*, *Cymbidium aloifolium*, and *Oberonia acaulis* were correlated towards lower elevation with host species like *Castanopsis indica* and *Engelhardia spicata*. *Bulbophyllum elatum*, *Bulbophyllum muscicola*, *Cymbidium × gammieanum*, *Liparis bootanensis*, *Otochilus fuscus*, and *Vandopsida undulata* had strong affinity towards characteristics of the host species like ever greenness, bark texture, height and diameter. *Dendrobium chryseum*, *Dendrobium moniliforme*, *Gastrochilus calceolaris*, *Otochilus lancilabius*, *Pleione praecox*, and *Pinalia spicata* were the species equally distributed towards all gradients with host like *Alnus nepalensis*, *Lyonia ovalifolia,* and *Rhododendron arboretum*.

## 3. Discussion

Orchids have maintained their allure long due to medicinal properties and aesthetic value in global horticulture and floriculture. The scientific exploration of orchids in Nepal has seen various studies over time [4,5,7,18,19,20,21,22] yet the investigation of the Tinchule Mountain region remains incomplete, and a comprehensive checklist of Nepal’s orchid flora, including distribution ranges, is still pending, with this study aiming to fill that gap by identifying about 18% of the total species [23,24,25]. Nepal’s unique phytogeography, bioclimatic zones, and diverse vegetation foster a rich diversity of flora, providing ideal conditions for the growth of native orchids and other vascular plants [7,26]. Our research highlights a distinctive pattern of epiphytic orchid species richness along an elevation, peaking at elevations of 2100 m to 2200 m. This study also revealed that the abundance of epiphytic orchids is directly linked to the number of host plants, with host species possessing broader geographical ranges supporting more orchid species, particularly at elevations of 2100 m to 2200 m. However, other studies have reported the areas at around 1500–1600 m, typically rich in epiphytic orchids where the climate is warm and humid [13,27,28,29], but also found to have fewer host species due to the prevalence of agricultural land and settlements [30].

The present study highlights the significance of host specificity in supporting the growth and abundance of epiphytic orchid species, revealing that the Fagaceae family (*Quercus lamellosa*, *Quercus semecarpifolia*, *Castanopsis indica*) serves as the most suitable host, followed by Lauraceae (*Cinnamomum glanduliferum*, and *Persea duthiei*), Ericaceae (*Rhododendron arboretum*, *Lyonia ovalifolia*), Betulaceae (*Alnus nepalensis*), Theaceae (*Schima wallichii*) and Daphniphyllaceae *(Daphniphyllum himalense*). Conversely, families Pinaceae *(Pinus roxburghii*, *Pinus wallichiana*), Taxaceae (*Taxas wallichiana*), and *Toricellia dumosa* were identified as the least favorable hosts, attributed to factors like resin presence and higher altitude distribution [8,29,31]. We recorded well developed *Bulbophyllum affine* in *Pinus roxburghii* and *Gastrochilus affinis* in *Toricellia dumosa*, and *Pinus wallichiana*. *Gastrochilus distichus*, *Oberonia falcate*, and *Vandopsida undulata* were also recorded from the *Berberis* species. Similarly, *Bulbophyllum roseopictum*, *Goodyera recurva*, *Pleione praecox*, and *Pleione humilis* were recorded from *Viburnum* species. The diversity of epiphytic orchids was strongly correlated with host characteristics such as basal area, height of the host tree, host type (deciduous or evergreen), bark texture, and vertical stratification, underscoring the importance of larger and older host species in providing ample substrate for orchid germination and colonization, thereby enhancing species diversity [13,29]. This study corroborates findings that species richness increases with the host larger size (DBH) and that epiphytic orchids thrive better on evergreen species due to consistent humidity levels. It was also found that orchids prefer larger older hosts with rough bark textures, which aid in moisture retention and seedling recruitment, and that the presence of lichen and moss on these hosts further supports orchid growth [32,33,34,35]. The study found the upper basal layer (L2) to have the highest orchid richness, likely due to lower disturbances than in the basal layers affected by human activities. Habitat-wise, forest areas were identified as the most conducive for epiphytic orchid richness, followed by mixed areas, agricultural lands, and settlements.

The research on epiphytic orchids demonstrates their dependency on a mix of physical environment factors, host characteristics, and ecological disturbances for their survival and proliferation. Key physical aspects include the orchids’ specific epiphytic habitats, influenced by factors such as latitude [27,36], altitude [7], light intensity [37], water availability and relative humidity [38,39]. A significant finding from CCA biplot analysis highlighted the correlation between species composition and environmental factors, notably altitude, which affects species turnover and diversity, particularly at higher elevations. The diversity of epiphytic orchid species is also shaped by the topography, climate, land use, and specific characteristics of their host species, such as type, surface roughness, and size [30,40]. The biogeographical range of each host species and orchid plays a crucial role, with certain topographies favoring specific host-orchid associations, indicating that the composition of epiphytic orchid communities is determined by a complex interplay of multiple environmental factors rather than a single element.

In Nepal, orchids span an extensive altitudinal range from the tropical lowlands at 60 m to the alpine regions at 5200 m, making them integral to the country’s rich biodiversity [41]. Orchids, used traditionally for medicinal, culinary, and decorative purposes [42,43], are faced extinction risks primarily due to their epiphytic nature, making them highly vulnerable to threats like illegal trade [17,44,45], deforestation, climate change, and urbanization. Recognizing these challenges, Nepal has implemented a robust framework of policies, acts, and regulations across various administrative levels to protect its biodiversity. The 2015 constitution underscores the government’s commitment to biodiversity, supported by several legislative and policy measures such as the National Parks and Wildlife Conservation Act (1973), the Forest Act (1993), and the Environmental Protection Act (1994), among others. Additionally, Nepal’s participation in international treaties, including the Ramsar Convention and the Convention on Biological Diversity, highlights its global commitment to conservation efforts. Despite these efforts, the issuance of a policy (2008), allowing the collection of wild orchids for trade without clear sustainable harvesting guidelines, has been counterproductive. To ensure effective conservation and sustainable use of orchids, it is critical to conduct studies on orchid population biology, establish distinct conservation and trade policies, and require Initial Environmental Examinations (IEEs) and Environmental Impact Assessments (EIAs) before permitting the harvesting of wild orchids. This study enriches our understanding of plant ecology and informs conservation practices critical for maintaining the health and diversity of Himalayan ecosystems.

## 4. Materials and Methods

### 4.1. Physiography of the Study Area

Tinchule Mountain is situated on the boundary of Kathekhola Rural Municipality-3 in the southeast and Tarakhola Rural Municipality-1 and 2 in the west, in the Baglung district. It also borders Beni Municipality-3 in the northeast, which is part of the Myagdi district. The ecological sampling region encompasses the areas of Kathekhola Rural Municipality-3 and Beni Municipality-3. The entire region falls within an elevation range of 1300 m to 2800 m. Based on the location and Universal Transverse Mercator (UTM) coordinates in Nepal, the examined region falls on latitude 28.18° to 28.35° N, longitude 83.47° to 83.52° E (Figure 1).

The study area falls within the bio-climatic region that spans from subtropical to temperate. The predominant vegetation in this area includes subtropical broad-leaved evergreen forest (comprising species such as *Schima wallichii*, *Castanopsis indica*, and *Alnus nepalensis*), subtropical pine forest (dominated by *Pinus roxburghii*), lower temperate mixed broad-leaved forest (consisting of *Laurel* species), upper temperate broad-leaved forest (featuring *Quercus semecarpefolia*, and *Quercus lamellosa*), upper temperate mixed broad-leaved forest (comprising *Aesculus*/*Juglans*), and temperate coniferous forest (with species like *Pinus wallichiana*, *Taxus wallichiana*, and *Tsuga dumosa*).

### 4.2. Field Sampling

To assess the variety, quantity, and affinity of orchids, we performed ecological sampling employing a methodical approach called belt transects. Within every horizontal transect, we set up six circular plots, each with a diameter of 5.6 m (≒ 100 m^2^). These plots were carefully placed, maintaining a minimum distance of 50 m between them and covering a 100 m elevation span in both study sites. We meticulously documented a range of environmental factors for each plot, including elevation, aspect, slope, and geographical coordinates. To accomplish this, we utilized specialized instruments including an altimeter, clinometer, compass, and GPS device.

Our sampling expedition began at an altitude of 1300 m in Varpokh and continued all the way to the Tinchule peak, which stands at 2800 m, tracing the southeastern slope in Baglung. At the same time, we conducted sampling from 1300 m in Marek up to the Tinchule peak on the northeast slope of Myagdi. In sum, we examined 192 plots across these two chosen locations, with 96 plots studied in each area. Within each designated plot, all tree species with a DBH of at least 10 cm were thoroughly examined to identify the presence of epiphytic orchids whereas in the case of shrub host species all the hosts are included. The investigation of epiphytic orchids within trees involved a modified methodology which included measuring tree height in meters (m) using a clinometer [16,17].

### 4.3. Analysis of Bark Textures and Canopy Layers of Host Trees, and Land Uses

The differences between bark texture categories, host types, canopy layers, and land uses were explored through observation. The height of the tallest canopy tree was used to visually identify different tree layers for the purpose of sampling epiphytic orchids. The categorization of tree height followed a systematic division, with the tallest trees divided into five equal sections, denoted as L1 to L5. These sections represented the lower basal layer (L1), upper basal layer (L2), middle layer (L3), lower canopy layer (L4), and upper canopy layer (L5). Trees shorter than the canopy height were examined in layers corresponding to their height relative to the canopy tree. For instance, trees below 10 m in height were categorized into two layers: L1 up to 5 m and L2 up to 10 m. The epiphytic orchids in each tree layer were documented separately, and individual orchids were counted at the putative genet level, comprising all ramets within a clump.

Bark rugosity or texture was categorized into three levels: smooth, represented by species like *Ficus neriifolia* and *Ilex dipyrena*; medium, as seen in *Juglans regia*, *Alnus nepalensis,* and *Aesculus indica*; and rough, akin to *Schima wallichii*, *Quercus* species, *Cornus capitata*, and *Rhododendron arboretum* [9]. Host types were classified into two categories: deciduous and evergreen. Land use was categorized into four types: forest area, agricultural land, settlement area, and mixed area.

The presence of orchid species within the observable range was directly recorded by the naked eye. Orchids growing beyond the visible range were studied by climbing trees using single rope climbing techniques [46] and aided by long-range binoculars [47]. The majority of orchid species and their host plants were distinguished in the natural environment by using pertinent local flora resources. To confirm the identity of these orchid species, various literature sources were consulted [24,25,26] in addition to examination of herbarium specimens stored at the National Herbarium and Plant Laboratories (KATH).

### 4.4. Statistical Analysis

We have prepared a map of the study area using map scale 1:30,000 via software ArcGIS pro 3.2 (ver. 2023) using the coordinates which were recorded during the field investigation. Additionally, we used the GraphPad Prism software (version 10.2.3, GraphPad Software, Boston, MA, USA, https://www.graphpad.com (accessed on 15 November 2024) to prepare the graph of the relation between species richness and elevation. Excel program was used to calculate the percentage of epiphytic orchid species abundance for each host family by dividing the total abundance of each host family by the sum of the total abundance for all host families.

Pearson’s correlation is employed to measure the relation between elevation, DBH, and height using the statistical software R version 4.0.5 (R Core Team 2018). Host preferences were analyzed based on the diversity and abundance of orchids, with hosts exhibiting the highest numbers of orchids considered most preferred. Further analysis examined the relationships between host-related variables and orchid diversity and abundance across various vertical layers. To assess the relative significance of physical and host-related variables in shaping the composition of epiphytic orchid species, we utilized CCA, presenting the fraction of variation in species data explained by each variable using a Monte Carlo Permutation test with 999 replicates (version 5.31, MjM Software Design, Gleneden Beach, OR, USA, 2006).

## 5. Conclusions

This study has brought to light a remarkable diversity of epiphytic orchid species in the Nepalese mountains, with significant variations in the composition of orchid communities, as well as species richness along the elevation. The composition of orchid populations is intricately linked to local factors such as vegetation and climate, as well as specific characteristics like the basal area, height, host type (deciduous or evergreen), and the bark texture of the host species and their respective families. The richness of epiphytic orchid species were increased as the number, size, height, and evergreen nature of host species grow. Among host families, the family Fagaceae exhibited the highest occupancy of epiphytic orchids, whereas the family Pinaceae displayed lower occupancy. Numerous studies focused on different epiphytic orchid species have highlighted the suitability of different elevations for their richness and abundance. To gain a comprehensive understanding of the broader patterns of epiphytic orchid diversity and its relationship with host species, it is imperative to conduct studies on a larger spatial scale, ranging from tropical to alpine regions. Given the lengthy propagation and reproductive processes of epiphytic orchids with host-specific manner, the initial step for their conservation will be the understanding of host-epiphytic orchid relationship within the biogeographic ecosystem.

## Figures and Tables

**Figure 1 plants-13-03256-f001:**
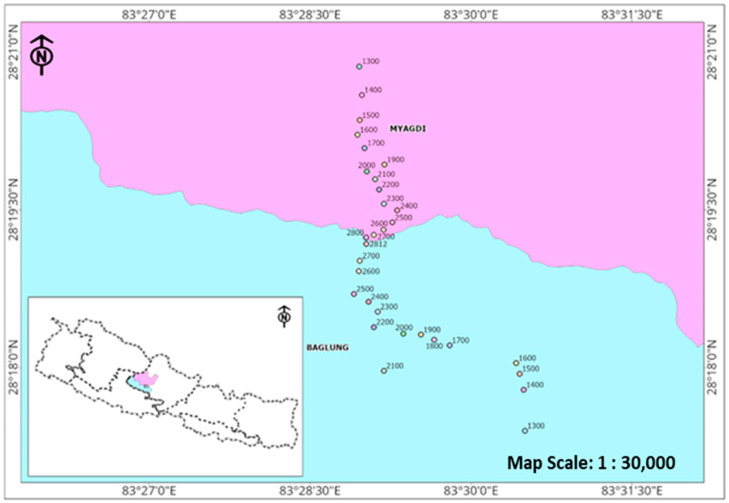
Map of Nepal showing study areas with study sites along the elevation. Examined plots are indicated with altitude (Baglung in blue and Myagdi in pink).

**Figure 2 plants-13-03256-f002:**
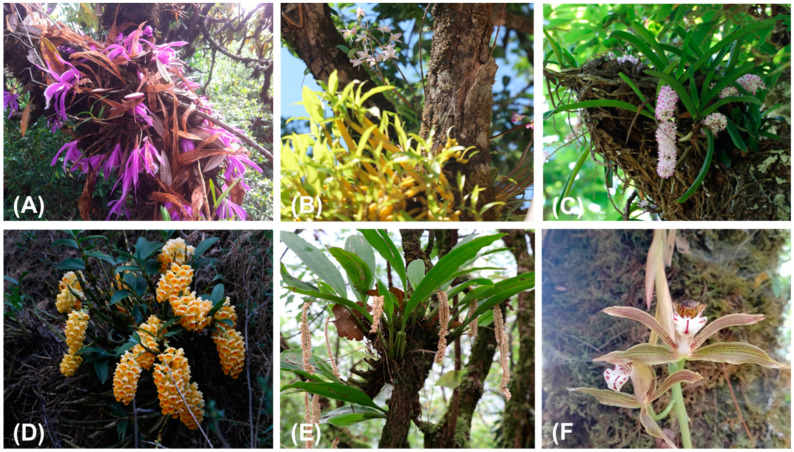
Some epiphytic orchid species observed in the Tinchule Mountain, Nepal. (**A**) *Pleione praecox*: found mainly on *Rhododendron arboreum*, (**B**) *Dendrobium amoenum*: recorded across all host layers, primarily in L3, L4, and L5 on *Alnus nepalensis*, (**C**) *Rhynchostylis retusa*: threatened due to its ornamental use, it grows mostly on deciduous and semi-deciduous trees like *Engelhardia spicata*, (**D**) *Dendrobium densiflorum*: abundant in the study area, especially in *Castonopsis* forests, (**E**) *Pholidota pallida*: found mainly on *Lyonia ovalifolia*, which has a highly rugose bark, and (**F**) *Cymbidium erythraeum*: prefers mossy trunks, particularly *Quercus semecarpifolia*.

**Figure 3 plants-13-03256-f003:**
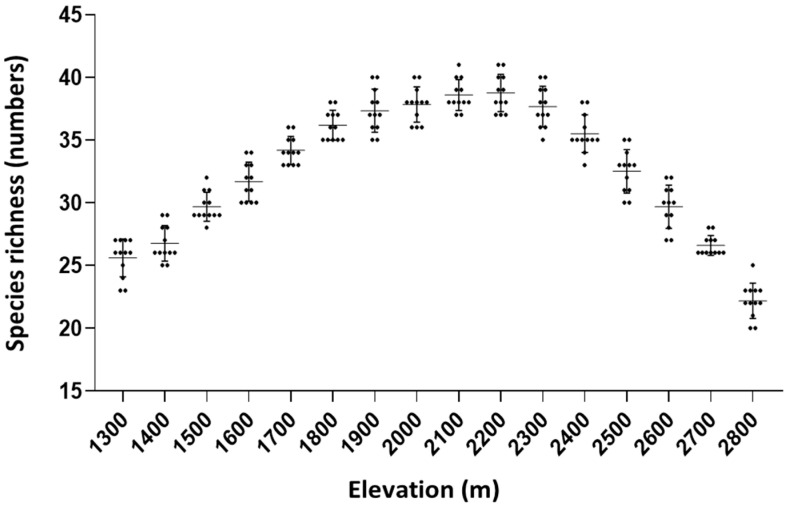
Species richness along the elevation of the study areas. Each dot indicates the observed number of orchid species in each examined plot. Horizontal and vertical lines indicate the mean richness values and standard deviations, respectively.

**Figure 4 plants-13-03256-f004:**
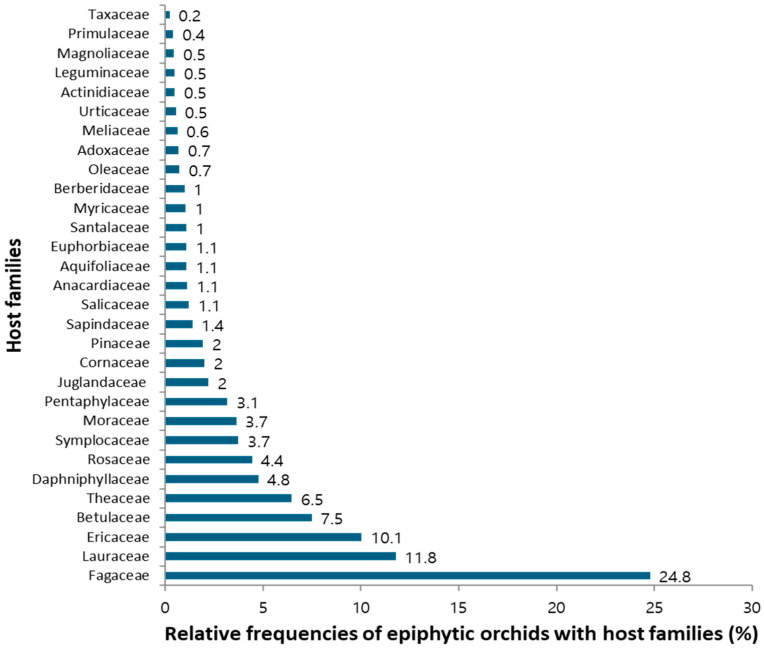
Relative frequencies of epiphytic orchids with host families.

**Figure 5 plants-13-03256-f005:**
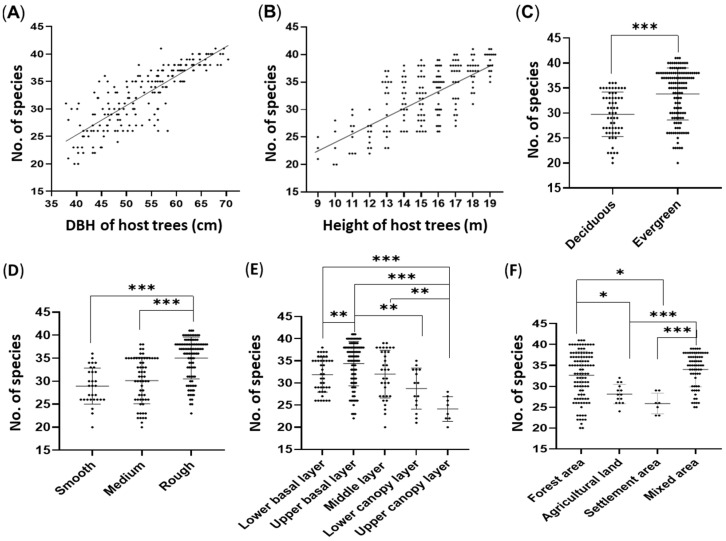
Relations between numbers of epiphytic orchid species with host properties and land uses: (**A**) diameter at breast height (DBH) of the host tree (cm), (**B**) height of the host species (m), (**C**) types of host species, (**D**) bark textures of the host species, (**E**) vertical layers of the host species, and (**F**) land uses where host species are grown (* *p* < 0.05; ** *p* < 0.01; and *** *p* < 0.001).

**Figure 6 plants-13-03256-f006:**
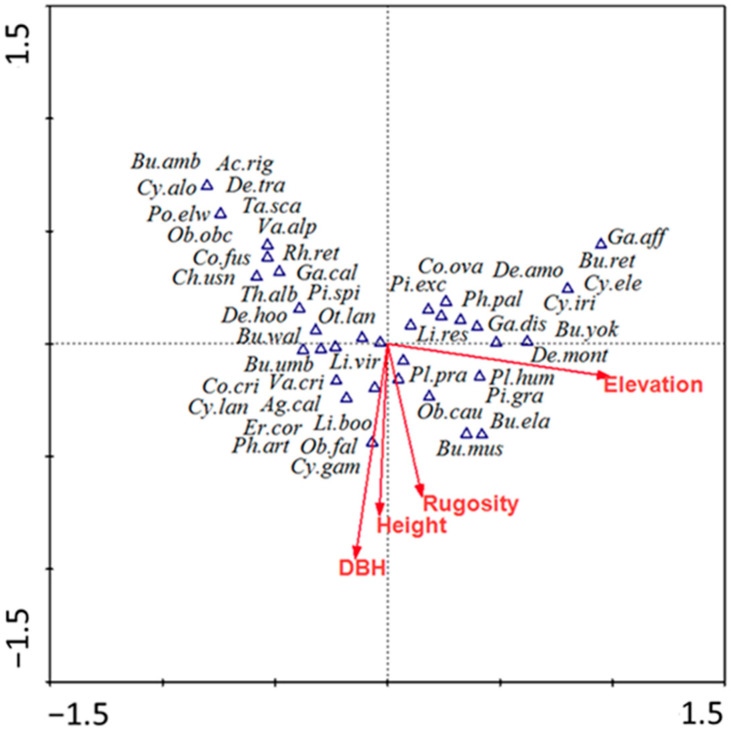
Canonical correspondence analysis showing the association between different epiphytic orchid species composition and predicted variables of elevation, diameter at breast height (DBH), height, and rugosity. The red lines show the predicted variables, where the blue triangles indicate the host plot.

**Table 1 plants-13-03256-t001:** Pearson’s correlation matrix (*r*) between the influencing factors for epiphytic orchids.

	Elevation	Host Trees	
		DBH ^1^	Height
Elevation	–		
DBH ^1^	−0.13 (*p* = 0.07)	–	
Height	−0.06 (*p* = 0.42)	0.74 (*p* < 0.001)	–

^1^ Diameter at breast height.

## Data Availability

The data presented in this study are available on request from the corresponding author.

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
