# Peer review of "Diversity Patterns of Epiphytic Orchids Along Elevation in the Mountains of Western Nepal"

_plants, 2024, doi:10.3390/plants13223256_

Round 1
Reviewer 1 Report
Comments and Suggestions for Authors
An interesting paper dealing with an important question - the species diversity of orchids along the elevation gradient, together with the correlations of their species diversity with other environmental factors, This analysis is quite comprehensive. I do not see any problems with either aspect of this research like the way of data collecting, sampling design or statistical analysis. Maybe the help of a native English speaker might improve the elegance of certain expressions. However, the paper is a good read even without that and can also be published as it is.
Author Response
An interesting paper dealing with an important question - the species diversity of orchids along the elevation gradient, together with the correlations of their species diversity with other environmental factors. This analysis is quite comprehensive. I do not see any problems with either aspect of this research like the way of data collecting, sampling design or statistical analysis. Maybe the help of a native English speaker might improve the elegance of certain expressions. However, the paper is a good read even without that and can also be published as it is.
Answer: Thank you for taking the time to review our manuscript and for your positive comments. We greatly appreciate your valuable feedback. Based on the reviewers’ recommendations, we have undertaken a series of comprehensive revisions to address each of concerns.
Reviewer 2 Report
Comments and Suggestions for Authors
Introduction
L. 34: I would say “usually called the orchid family”
L. 44: here you could add a few lines about the orchid family in general and how the distribution of orchids (both terrestrial and epiphytic) is driven by specific factors and among them altitude. There are several papers published so far referring to terrestrial and epiphytic orchids to cite.
L. 52: Before that paragraph, I would like to see a paragraph about the conservation challenges the orchids face. After that, you can be more specific about Kathmandu Valley.
Material and methods
L.286-287: you say that the plots were strategically positioned. What does that mean? Besides the minimum distance between two plots (set to be more than 50 m) did you follow any other restriction? Can you characterize the selection of plots as random? Moreover, I would like to ask why did you select 6 plots at each 100-m altitudinal interval and not 4 or 7-8? Did you follow a specific sampling protocol?
L. 327: …… the coordinates which were recorded ….
L. 328: ….. we used …..
L.334-335: “between elevation, DBH and height”. Add a comma after elevation
L. 342: “detrended correspondence analysis (DCA)” and not “detrended DCA”
Results
L. 91-92: do you mean individuals instead of species?
L.99-101: this is wrong! It is not the host tree species that are positively associated with species richness but the DBH of the host tree species. At least this is what Fig. 5A shows.
L.101-102: “significant positive association (R² = 0.51) with species richness (Figure 5B)”.
L.108-143: The methodology of exploring the differences between bark texture categories, canopy layers and land uses is not referred to the material and method section. I think that as the number of categories is larger than 2, authors should use the Kruskal-Wallis test and state the results of these tests accordingly.
L. 144: showed instead of shows
L. 151: which datasets? Nothing was specified in the material and methods
L. 151-164: I am not convinced about the usefulness of DCA and CCA! Alternatively, you could use the much better for this kind of data outlying mean index analysis (OMI). However, I do not think that the analysis of the orchid communities was among the aims of your study. The analysis of the communities and the affinity of orchids with host trees could be explored using OMI and Indicator Species Analysis (ISA). These could be excluded as are not among the aims of your study and write another paper exploring all these.
Figure 3: What do dots mean?
Discussion
L. 190-194: I do not think that the knowledge of the epiphytic flora of Nepal was among the aims of this study! You didn’t perform a detailed survey to fill that gap! Here you should be focused on the significance of these studies (diversity patterns~elevation) and how their outcomes could be used.
L. 217-229: and what does that mean? I would like to read about the necessity of retaining large sized trees in Nepal because in this way epiphytic orchids could be conserved.
Comments on the Quality of English LanguageSome parts of the manuscript are very well written but others not.
Author Response
L. 34: I would say “usually called the orchid family”
Answer: Thank you very much for taking the time to review our manuscript. We have revised in the line 34 as “The Orchid family is an extensive and diverse group of flowering plants, encompassing approximately 28,000 species distributed worldwide”.
L. 44: here you could add a few lines about the orchid family in general and how the distribution of orchids (both terrestrial and epiphytic) is driven by specific factors and among them altitude. There are several papers published so far referring to terrestrial and epiphytic orchids to cite.
Answer: Thank you very much for this comment. We added following paragraph as the comment: In Nepal, the distribution of orchids, particularly terrestrial and epiphytic types, is influenced significantly by the country's diverse topography and altitude gradient. Higher altitudes with cooler, humid microclimates favor terrestrial orchids, while lower, humid regions with dense canopy cover support epiphytic orchids that thrive in moist, filtered light conditions provided by their hosts. The relation between altitude and orchid diversity, showing that epiphytic orchids are commonly found at lower altitudes (500 – 1,500 m), where warmth and moisture are abundant, while terrestrial orchids are more prevalent at mid-to-high altitudes. Altitude, canopy cover, and microclimate as key factors in orchid distribution, showing species-specific adaptations to different altitudinal ranges [4,7,9,13].
L. 52: Before that paragraph, I would like to see a paragraph about the conservation challenges the orchids face. After that, you can be more specific about Kathmandu Valley.
Answer: Thank you for the comment. We have added a paragraph about your comment: In the Kathmandu Valley, orchids face significant conservation challenges, primarily due to urbanization, deforestation, and habitat fragmentation. As the valley continues to urbanize, the natural habitats of orchids are increasingly threatened by infrastructure development and agricultural expansion [14]. Additionally, illegal collection of orchids for ornamental and medicinal purposes, along with climate change, has further exacerbated the vulnerability of these plants [15]. Conservation efforts in the region need to focus on habitat protection, sustainable management, and raising awareness about the ecological value of orchids to ensure their survival in this rapidly changing environment.
Material and methods
L. 286-287: you say that the plots were strategically positioned. What does that mean? Besides the minimum distance between two plots (set to be more than 50 m) did you follow any other restriction? Can you characterize the selection of plots as random? Moreover, I would like to ask why did you select 6 plots at each 100-m altitudinal interval and not 4 or 7-8? Did you follow a specific sampling protocol?
Answer: Thank you very much for this comment. "strategically positioned" was an inappropriate choice of words. In our study area, selecting six plots at each 100-meter altitudinal interval was deemed appropriate based on the inclined/sloped conditions of the study area. We did not choose four or seven to eight plots because six provided a balanced representation without any other underlying reason. The sentence has been revised: These plots were carefully placed, maintaining a minimum distance of 50 m between them and covering a 100 m elevation range at both study sites.
L. 327: …… the coordinates which were recorded ….
L. 328: ….. we used …..
L. 334-335: “between elevation, DBH and height”. Add a comma after elevation
L. 342: “detrended correspondence analysis (DCA)” and not “detrended DCA”
L. 101-102: “significant positive association (R² = 0.51) with species richness (Figure 5B)”.
L. 144: showed instead of shows
Answer: Thank you very much for your comments. The manuscript was revised according to all the comments.
L. 91-92: do you mean individuals instead of species?
Answer: Thank you for the comment. Here, we counted ramets, as orchids are found in bunches. Therefore, we have considered each ramet as one.
L. 99-101: this is wrong! It is not the host tree species that are positively associated with species richness but the DBH of the host tree species. At least this is what Fig. 5A shows.
Answer: Thank you very much for the comment. Firstly, the DBH of host species showed a strong positive correlation (R² = 0.69) with the richness of epiphytic orchids (Figure 5A) is added.
L. 108-143: The methodology of exploring the differences between bark texture categories, canopy layers and land uses is not referred to the material and method section.
Answer: Thank you very much for the comment. “The differences between bark texture categories, canopy layers, and land uses were explored through observation.” Is added in the materials and method section.
L. 151: which datasets? Nothing was specified in the material and methods
L. 151-164: I am not convinced about the usefulness of DCA and CCA! Alternatively, you could use the much better for this kind of data outlying mean index analysis (OMI). However, I do not think that the analysis of the orchid communities was among the aims of your study. The analysis of the communities and the affinity of orchids with host trees could be explored using OMI and Indicator Species Analysis (ISA). These could be excluded as are not among the aims of your study and write another paper exploring all these.
Answer: Thank you for the comments and suggestions. We aimed to provide comprehensive information in our research, including orchid communities which may aid in their exploration, identification, and conservation. We appreciate your feedback, thus, the corresponding descriptions (Results, Materials and Methods sections) and Table 2 were completely deleted from the manuscript.
Figure 3: What do dots mean?
Answer: Thank you for your comment. The word 'dots' means observed numbers of orchid species per examined plot. The following sentence was added to the figure caption: Each dot indicates observed number of orchid species in each examined plot.
L. 190-194: I do not think that the knowledge of the epiphytic flora of Nepal was among the aims of this study! You didn’t perform a detailed survey to fill that gap! Here you should be focused on the significance of these studies (diversity patterns~elevation) and how their outcomes could be used.
Answer: Thank you for your comment and suggestions. In our study, we documented around 18% of the total orchid species found in Nepal, which we believe adds value to understanding the diversity patterns in relation to elevation. Therefore, we tried to include the significance of our observations and to demonstrate how such data can be useful for further research and conservation planning.
Reviewer 3 Report
Comments and Suggestions for Authors
I am grateful to have had the opportunity to read the article, entitled "Diversity Patterns of Epiphytic Orchids along Elevation in the Mountains of Western Nepal," which was submitted to the academic journal Plants (MDPI). The title of the paper indicates that it is a comprehensive study involving a substantial amount of data. However, the manuscript in question comprises merely a brief communication regarding preliminary research conducted in a limited geographical area: specifically, the Khathkhola-3 region in Baglung and the Beni-3 area in Myagdi, both located within Nepal (see lines 60–61). Unfortunately, the manuscript does not meet the requisite standards for publication in its current form. The introduction requires substantial revision and expansion. The section presents data at the level of the school, rather than data intended for use by professionals. The individual paragraphs are inconsistent and contain an inconsistent set of data, making it difficult to determine the study's most important findings. The results chapter is competently written, although the presented results themselves are not particularly noteworthy. It is anticipated that the discussion chapter will offer commentary on the results obtained, which also necessitates comprehensive improvement in this regard. It is unclear whether the initial sentence of this chapter (lines 189-191) is necessary. The data presented in lines 209-217 are, in fact, the results of the study, rather than a discussion of them. Moreover, the conclusions presented in this article are unduly general and do not align with the specific content of the work. It is recommended that the title be modified to be more precise to more accurately reflect the content of the work. The presented results are limited to the research area and may not be representative of all of Western Nepal. It is also recommended that this work be presented as a communication, rather than a research article, given that the data presented lack significant scientific value. For those with a familiarity with Nepalese flora, the findings of this research are not unexpected and align with existing knowledge on the subject matter.
Comments on the Quality of English LanguageThe manuscript should be thoroughly checked for English language and writing errors.
Author Response
I am grateful to have had the opportunity to read the article, entitled "Diversity Patterns of Epiphytic Orchids along Elevation in the Mountains of Western Nepal," which was submitted to the academic journal Plants (MDPI). The title of the paper indicates that it is a comprehensive study involving a substantial amount of data. However, the manuscript in question comprises merely a brief communication regarding preliminary research conducted in a limited geographical area: specifically, the Khathkhola-3 region in Baglung and the Beni-3 area in Myagdi, both located within Nepal (see lines 60–61). Unfortunately, the manuscript does not meet the requisite standards for publication in its current form. The introduction requires substantial revision and expansion. The section presents data at the level of the school, rather than data intended for use by professionals. The individual paragraphs are inconsistent and contain an inconsistent set of data, making it difficult to determine the study's most important findings. The results chapter is competently written, although the presented results themselves are not particularly noteworthy. It is anticipated that the discussion chapter will offer commentary on the results obtained, which also necessitates comprehensive improvement in this regard. It is unclear whether the initial sentence of this chapter (lines 189-191) is necessary. The data presented in lines 209-217 are, in fact, the results of the study, rather than a discussion of them. Moreover, the conclusions presented in this article are unduly general and do not align with the specific content of the work. It is recommended that the title be modified to be more precise to more accurately reflect the content of the work. The presented results are limited to the research area and may not be representative of all of Western Nepal. It is also recommended that this work be presented as a communication, rather than a research article, given that the data presented lack significant scientific value. For those with a familiarity with Nepalese flora, the findings of this research are not unexpected and align with existing knowledge on the subject matter.
Answer: Thank you very much for your thorough review and valuable feedback. We are grateful for your detailed comments, which have greatly contributed to improving the quality and rigor of our manuscript. Based on your recommendations, we have undertaken a series of comprehensive revisions to address each of your concerns.
To enhance the introductory section, we expanded it significantly, providing additional context on the ecological significance of epiphytic orchids, as well as a clearer explanation of the study's objectives and methodology. This has been done to ensure that the background and rationale for the study are more appropriately framed for an academic audience, and to present the preliminary findings in a manner that aligns with the standards of the journal.
In the results section, we carefully reviewed and refined our presentation to highlight the key findings and ensure that they are communicated with greater clarity and relevance. We have also restructured the discussion section to offer a deeper analysis and interpretation of the results, as well as their broader implications for biodiversity patterns and conservation in the region. We removed any extraneous or redundant information and focused on strengthening the logical flow of ideas, to enhance the readability and academic rigor of this section.
Round 2
Reviewer 2 Report
Comments and Suggestions for Authors
none
Reviewer 3 Report
Comments and Suggestions for Authors
The article sent for re-review has not been thoroughly changed and supplemented, minor corrections have been made. I still think that this paper is very weak scientifically. The added paragraphs on the distribution of orchids in Nepal (lines 54-71) are very general and do not lead to much in general. For the paper to be cited by other authors, it should contain valuable content; in my opinion, this paper can be an introduction to professional research in the future, but in its current form it is not suitable for publication.